# Cavitation Peening: A Review

**Hitoshi Soyama** 

Department of Finemechanics, Tohoku University, Sendai 980-8579, Japan; soyama@mm.mech.tohoku.ac.jp;
Tel.: +81-22-795-6891; Fax: +81-22-795-3758

**Abstract:** The most popular surface modification technology used to enhance the mechanical properties of metallic materials is shot peening. Shot peening improves fatigue life and strength by introducing local plastic deformation pits. However, the pits increase surface roughness, which is a disadvantage for fatigue properties. Recently, cavitation peening, in which cavitation bubble collapse impacts are used, has been developed as an advanced surface modification technology. The advantage of cavitation peening is the lesser increase in surface roughness compared with shot peening, as no solid collisions occur in cavitation peening. In conventional cavitation peening, cavitation is generated by injecting a high-speed water jet into water. However, cavitation peening is different from water jet peening, in which water column impacts are used. In the present review, to avoid confusing cavitation peening and water jet peening, fundamentals and mechanisms of cavitation peening are described in comparison to water jet peening, and the effects and applications of cavitation peening are reviewed compared with the other peening methods.

**Keywords:** surface modification; cavitation peening; shot peening; water jet; pulse laser; ultrasonic

## 1. Introduction

As cavitation causes severe damage to hydraulic machinery such as pumps [1,2], cavitation is harmful to hydraulic components. Impact at cavitation bubble collapse can be used to enhance material properties, similar to shot peening. A peening method using cavitation impact is called "cavitation peening" [3] or "cavitation shotless peening" [4], as shots are not required in cavitation peening. The merit of cavitation peening is that the increase in surface roughness is less than that of conventional shot peening, as no solid collisions occur. In the case of cavitation peening, the impacts at bubble collapses are used; thus, understanding the mechanism of cavitation is important to enhance material properties without erosion.

Although cavitation impacts cause erosion in hydraulic components, cavitation peening can treat metallic materials without causing erosion, as cavitation peening finishes the treatment within the incubation period. The cavitation erosion, i.e., mass loss induced by cavitation, changes with exposure time to cavitation, and is classified into four stages: incubation stage, acceleration, maximum rate, and deceleration [5–7]. In the incubation stage, the cavitation impacts that affect the materials produce plastic deformation without mass loss. Note that the threshold level of each material can be obtained experimentally [8]. After the incubation period, the mass loss starts due to a fatigue fracture process, and the mass loss rate increases with time; thus, it is called the acceleration stage. As the optimum processing time of cavitation peening is about 1/25 to 1/5 of incubation period, no mass loss occurs on the treated surface [9]. After the acceleration stage, the mass loss rate reaches a maximum in the maximum rate stage, which then decreases in the deceleration stage.

At the beginning, cavitation peening was developed to mitigate stress corrosion cracking in nuclear power plants by reducing the tensile residual stress of subsurface of stainless steel [10]; it was successfully applied in the plants [11]. After the aggressive intensity of cavitation peening was

enhanced by optimizing conditions [4,12], the fatigue strength of mechanical components such as gears improved [13–15]. In the present review, the aggressive intensity of cavitation peening is discussed in terms of the cavitation peening results, such as improvement in fatigue strength, work hardening, and introduction of compressive residual stress. Recently, cavitation peening with abrasive particles was proposed to enhance the fatigue properties of additive manufactured titanium alloy by smoothing the surface roughness by introducing compressive residual stress into the subsurface [16]. The history of cavitation peening is described in Section 2.

In commonly used cavitation peening, cavitation is generated by injecting a high-speed water jet into a water-filled chamber. Although the water jet is used for cavitation peening, the mechanism of cavitation peening is different from that of water jet peening, in which shots accelerated by a water jet or water column impacts at the jet center are used [17–20]. Note that impact at cavitation bubble collapse is used in cavitation peening as mentioned above. The difference between cavitation peening and water jet peening is described in Section 6, and the classification map between cavitation peening region and water jet peening region is introduced in that section.

In this review, to efficiently expand applications of cavitation peening with reliability and safety, the fundamentals and applications of cavitation peening are described compared with other peening techniques.

## 2. History of Cavitation Peening

To mitigate the stress corrosion cracking of pressure vessels in nuclear power plants, mechanical surface treatments to reduce tensile residual stress in the subsurface in the submerged condition were investigated. At that time, peening methods using a submerged water jet [17] and a submerged pulse laser [21] were proposed. Both methods were successfully applied to nuclear power plants [11,22].

When a high-speed water jet is injected into water through a nozzle or an orifice, cavitation bubbles are generated inside or at the exit of the throat. The submerged water jet with cavitation bubbles is called a cavitating jet. Impulse pressure caused by a submerged high-speed water has two peaks that change with distance from nozzle [23]. When the residual stress on the impinging surface of stainless steel exposed to the cavitating jet was measured changing the standoff distance from the nozzle to the impinging surface, the compressive residual stress showed two peaks as a function of the standoff distance [10]. The peak at the near side of the nozzle was generated by impinging water columns in the jet center, i.e., water jet peening. The peak on the far side of the nozzle was caused by impacts of cavitation bubble collapses, i.e., cavitation peening.

Cutting and testing of materials using a simple cavitating nozzle was proposed [24], and material properties in a cavitation tunnel were improved by cavitation impact [25]. The residual stress on the surface of metal powders was changed by ultrasonic cavitation [26]. However, the treatment area using cavitation impacts is limited when cavitation is produced by cavitation tunnel and/or ultrasonic cavitation. Cavitating jets are suitable for controlling the treatment area and for aggressive intensity cavitation peening.

During the initial stage of cavitation peening development, cavitation is generated by injecting a high-speed water jet into water as mentioned above. In the case of a submerged jet, i.e., a cavitating jet in water, treating outer surfaces of tanks and/or pipelines is difficult. Soyama realized "a cavitating jet in air" by injecting a high-speed water jet into a low speed water jet, which was injected into air without a water-filled chamber, and demonstrated the introduction of compressive residual stress into a metallic surface and the resulting improvement of fatigue strength of stainless steel [27,28] and nitrocarburized steel [29]. The cavitating jet in air research has been followed by Marcon et al. [30,31]. The differences produced by the introduced distribution of compressive residuals stress between a cavitating jet in water and a cavitating jet in air is summarized in Section 4.1.

The main purpose of cavitation peening during the initial stage of development was mitigation of stress corrosion cracking in stainless steel, as mentioned above. In the second stage, cavitation peening was applied to enhance the fatigue properties of metallic materials such as silicon manganese

steel [32], aluminum alloy [4,33,34], carbonized chrome molybdenum alloy steel [12], and carbon steel [35] by improving the aggressive intensity of cavitation peening. In this review, the aggressive intensity of the cavitating jet is the ability of the cavitating jet to introduce compressive residual stress, arc height of treated plate, and the erosion rate. To enhance the aggressive intensity of the cavitating jet, a pressurized chamber was used to optimize the cavitating condition considering cavitation number (see Section 4.5). The fatigue strength of actual mechanical components, such as gears [14,36,37] and continuous valuable transmission (CVT) elements [13], was improved. Cavitation peening using a pressurized chamber improves valuable equipment, such as biomedical implants [38].

During the third stage of cavitation peening, to enhance aggressive intensity of the cavitating jet, the nozzle was optimized considering the outlet geometry of the nozzle throat [39]. The outlet bore of the nozzle throat was optimized [40], thereby enhancing the aggressive intensity of the jet by about 20 times. Then, a cavitator was proposed to supply cavitation nuclei into the cavitating jet [41] considering effect of the cavitator on severe cavitation erosion in a centrifugal pump [42]. As the vortices are also important, a guide pipe was placed at the downstream of the nozzle and optimized [41]. The cavitator and guide pipe enhanced each enhanced the aggressive intensity two-fold. Thus, the aggressive intensity of the jet was enhanced by $20 \times 2 \times 2 = 80$ times compared with the first stage of cavitation peening development, as shown in Figure 1. Recently, water flow holes near the nozzle outlet were proposed, and the optimized holes enhanced aggressive intensity by about 34% compared to without holes [43]. The details of effect of nozzle geometry is described in Section 4.3.

During the fourth stage of cavitation peening development, abrasive cavitation peening was proposed [16]. Additive manufactured (AM) metals are attractive materials as the components are directly manufactured from computer-aided design (CAD) data, along with other advantages. However, the limitations of AM metals are the fatigue life and strength due to the rough surface resulting from unmelted metallic particles. Although cavitation peening can improve the fatigue strength of AM metals drastically [44,45], the surface roughness is barely changed. Then, to create a smooth surface by abrasive collision, abrasion was added to the cavitating jet, improving the fatigue strength of titanium alloy manufactured by electron beam melting (EBM) [16]. The polishing inside of holes made by additive manufacturing was also proposed using abrasion and cavitation [46].

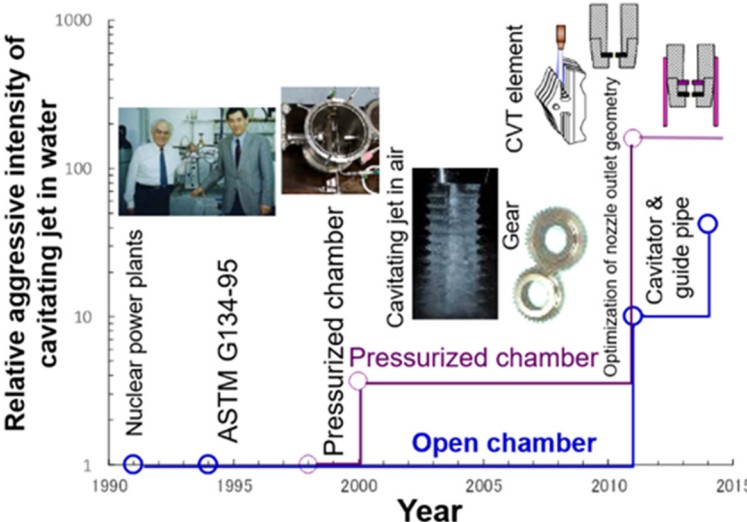

**Figure 1.** Development of relative aggressive intensity of cavitating jet, changing over time.

## 3. Cavitation

### 3.1. What Is Cavitation?

Cavitation is a phase change phenomenon from liquid to gas by decreasing pressure due to increase in velocity [2]. As shown in the Bernoulli equation (Equation (1)), when velocity $v$ increases, pressure $p$ decreases. Thus, when pressure reaches the vapor pressure of liquid $p_v$, the liquid becomes vapor, i.e., gas phase:

$$\frac{1}{2} \rho_L \, v^2 + p = const \tag{1}$$

where $\rho_L$ is the density of the liquid.

The most important parameter of cavitating flow is cavitation number $\sigma$ [2], which is defined by Equation (2). $\sigma$ is a ratio of dynamic pressure defined by the velocity and the static pressure considering $p_v$:

$$\sigma = \frac{p - p_v}{\frac{1}{2}\rho_L v^2} = \frac{p_2 - p_v}{p_1 - p_2} \approx \frac{p_2}{p_1} \tag{2}$$

where $p_1$ and $p_2$ are the upstream and downstream pressure of the orifice or nozzle, respectively, and are absolute pressures. In the case of a cavitating jet, $\sigma$ can be simplified using Equation (2) because $p_1 >> p_2 >> p_v$.

Figure 2 illustrates typical aspect of cavitation. In Figure 2, the water flows from the left- to the right-hand side through a Venturi tube [47]. In the narrow region where the pressure decreases due to the increase in flow velocity, water becomes cavitation bubbles, which are shown as white bubbles. In the expanded region where the pressure increases due to the decrease in flow velocity, cavitation collapses. This means that the gas phase becomes the liquid phase. As shown in Figure 2, in the cavitation collapsing region, the string-like cavitation bubbles are observed. In previous studies [1,2], cavitation caused severe erosion through vortex cavitation, as shown in Figure 3 [48]. This vortex cavitation consists of small tiny bubbles. Numerical simulation showed that cloud cavitation, consisting of small tiny bubbles, generates larger shockwaves comparing with a single bubble [49]. Thus, from the cavitation peening viewpoint, the generation of vortex cavitation consisting of tiny bubbles is important for enhancing peening intensity.

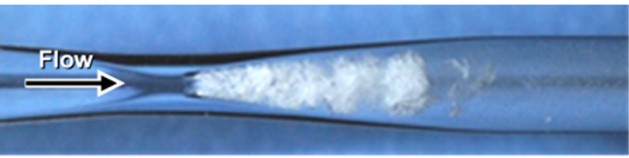

**Figure 2.** Aspect of cavitation through Venturi tube [47].

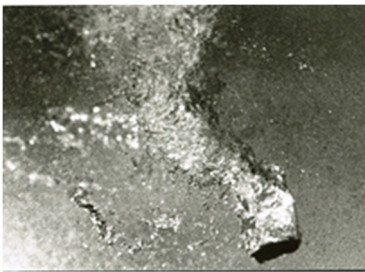

**Figure 3.** Aspect of vortex cavitation arising downstream of a butterfly valve [48].

Impact at cavitation bubble collapse can be generated by two methods [2]. This first is a microjet, which was observed numerically [50] and experimentally [51,52]. The far side of the bubble from the solid surface deforms, and the vapor–liquid interfaces becomes a microjet. Then, the jet hits the surface.

Note that a microjet in vortex cavitation was also observed experimentally [53]. The other method is the shockwave method, created after cavitation shrinking then rebounding due to gas pressure inside the bubble. At the rebound, i.e., expansion, a shockwave is generated. Figure 4 reveals the collapse of a bubble that was induced by a pulse laser using photo elasticity [54]; the behavior of a bubble induced by a pulse laser is similar to that of a cavitation bubble [52,55]. In Figure 4, the upper part is water and lower part is acrylic resin with polarization plates. The aspect was observed by a high-speed video camera. The shrunken bubble is shown in black shadow, as the light source was placed on the other side of the camera. At time (*t*) = 0 μs, the bubble size was the smallest and increased at *t* > 0 μs due to the rebound of the bubble. At *t* > 0 μs, the pressure wave was visualized in a black and white pattern due to photo elasticity. This result shows that the impact was produced at *t* = 0 μs, and the pressure wave was propagated in the target material.

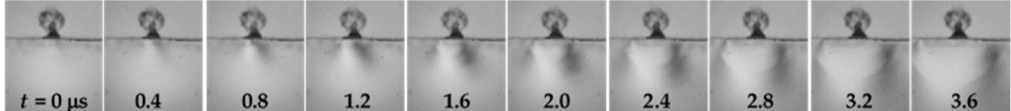

**Figure 4.** Visualization of impact at cavitation collapse by using photo elasticity [54].

### 3.2. How To Generate Cavitation

#### 3.2.1. Hydraulic Cavitation

The most common method used to generate cavitation for cavitation peening is hydraulic cavitation. This means that cavitation is created using a hydrodynamic phenomenon in a narrow flow passage such as a Venturi tube, orifice, and/or nozzle. As shown in Figure 2, cavitation is generated in the narrow region of a Venturi tube, and cavitation can be taken out from the tube by cutting the tube, as shown in Figure 5a [54]. The step type nozzle also creates cavitation inside the nozzle and near nozzle exit, as shown in Figure 5b [54]. Figure 6 shows a schematic diagram of a cavitating jet, a cavitating jet observed with a flush lamp, and the jet observed with a normal light source. To reveal the unsteady periodical aspect of the impinging cavitating jet, Figure 7 shows the impinging jet observed using a high-speed video camera. High-speed observation of a free cavitating jet and a impinging cavitating jet using a high-speed video camera was reported [56], and more details of the impinging jet were provided [57]. The condition of the cavitation jet in Figure 6b,c and Figure 7 was as follows; the injection pressure $p_1$ was 30 MPa, the downstream of the nozzle was at atmospheric pressure, i.e., 0.1 MPa, the nozzle diameter *d* was 2 mm, and the standoff distance from the upstream corner of the nozzle to the target was 262 mm. When a high-speed water jet is injected into water, cavitation is generated in the vortex core in the shear layer around the jet, where the pressure is lower, as shown in Figure 5b [58,59]. The vortex cavitations combine with each other, creating a big cloud of cavitation, which consists of tiny bubbles, as shown in Figures 6b and 7. When the cloud cavitation reaches the impinging surface, the cloud cavitation becomes a ring vortex cavitation, then part of ring cavitation collapses, producing an impact in a ring region on the target surface. This is why the typical treatment is performed by a fixed cavitating jet, as shown in Figure 8 [3]. In Figure 8, an aluminum specimen is used to clearly show a peening area. Note that when a free and/or impinging cavitating jet is observed using a normal continuous light source with a normal camera, it is very difficult to see the structure of the cavitating jet, as the jet was observed in Figure 6c. Cloud cavitation and/or ring vortex cavitation cannot be observed by such conventional observation. A figure in certain studies [60–62] showed that the vortex cavitation in the shear layer directly produces a ring region, but this is incorrect. Actually, the cloud cavitation behavior is important for optimizing cavitation peening, as larger cloud cavitation has a larger impact, which is used for cavitation peening [41], and the size of cloud cavitation is closely related to the frequency of the cloud shedding [41]. The similarity law of the shedding frequency was previously reported [63].

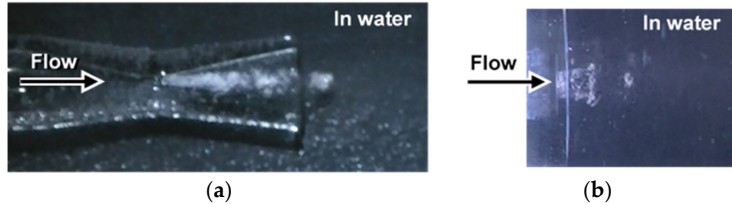

**Figure 5.** Aspect of cavitation through a nozzle [54]: (**a**) Venturi-type nozzle and (**b**) step-type nozzle.

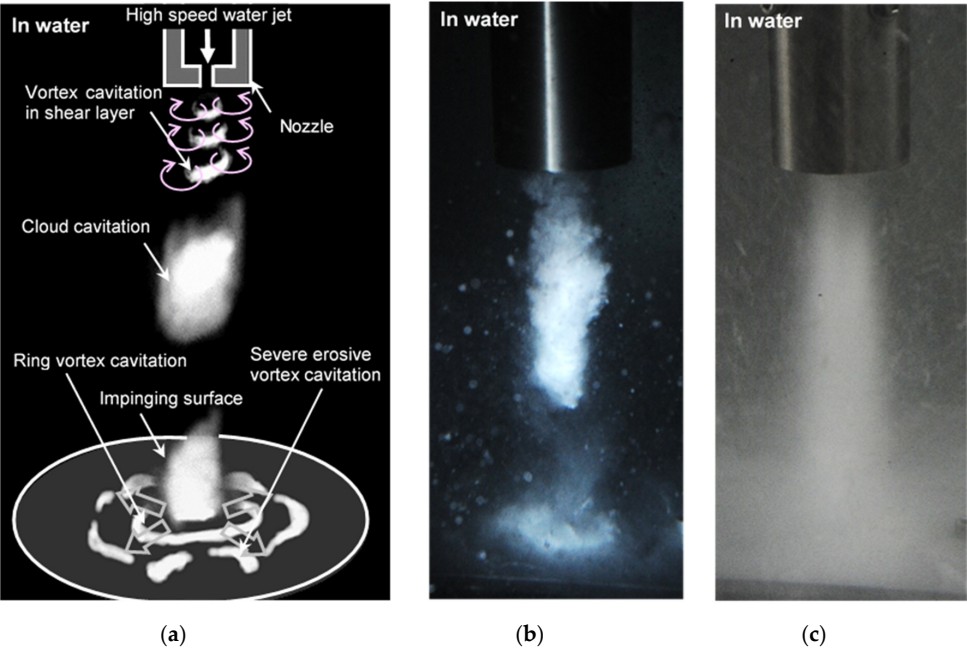

**Figure 6.** Aspects of submerged high-speed water jets, i.e., cavitating jet. (**a**) Schematic diagram; (**b**) Observation with flush lamp; (**c**) Observation with normal light.

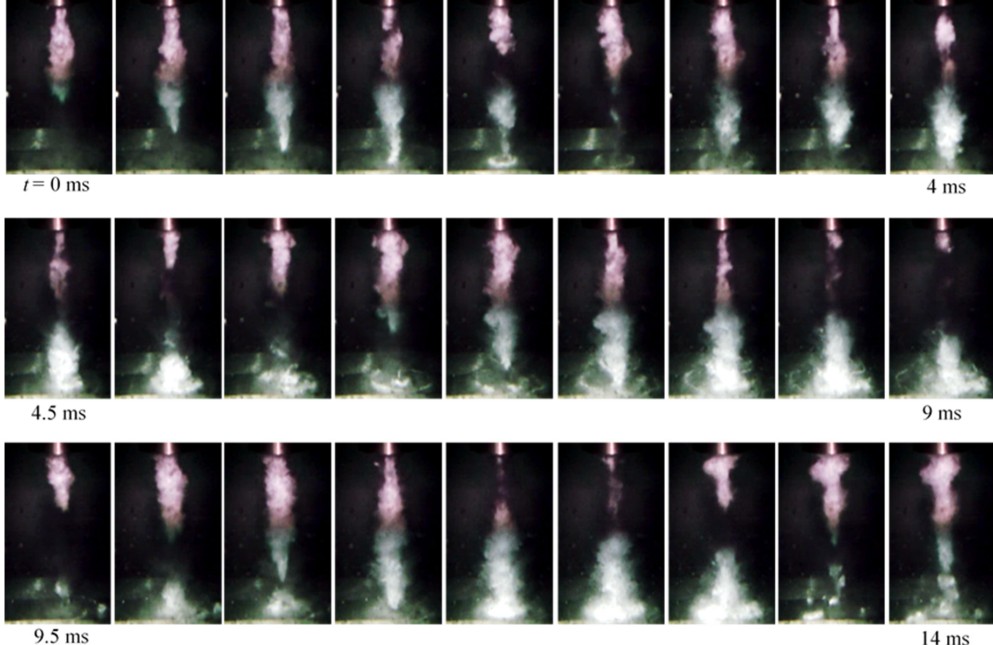

**Figure 7.** Aspect of impinging cavitating jet observed by high-speed video camera.

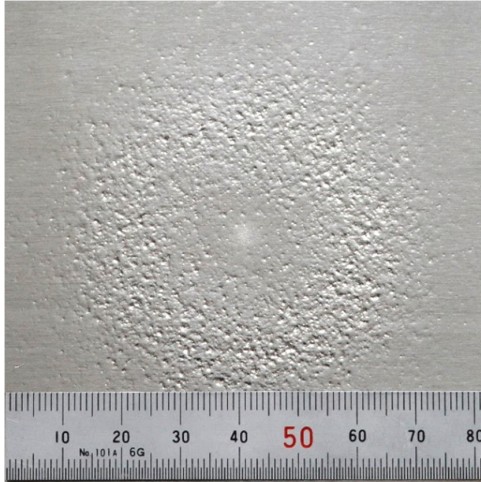

**Figure 8.** Typical treatment area by a fixed cavitating jet (pure aluminum, nozzle diameter $d$ = 2 mm, upstream pressure of nozzle $p_1$ = 30 MPa, downstream pressure of nozzle $p_2$ = 0.1 MPa, standoff distance $s$ = 262 mm, exposure time $t$ = 1 min) [3].

As mentioned above, a submerged high-speed water jet with cavitation, i.e., a cavitating jet in water, is used for conventional cavitation peening. The target should be placed in a water-filled chamber. Thus, treating the outside surface of tanks and pipelines, which are required to be treated by cavitation peening, is difficult. Then, Soyama realized a cavitating jet in air by injecting a high-speed water jet into a low-speed water jet, which was injected into air without a water-filled chamber, and demonstrated the introduction of compressive residual stress into metallic materials [27,64,65]. Typical aspects of a cavitating jet in water and in air are shown in Figure 9 [54]. As the periodical shedding of the cloud cavitation of the cavitating jet synchronizes with the velocity of the low-speed water jet under optimum conditions [64], the surface of the low-speed water jet shows a drastic wavy pattern, as shown in Figure 9. The improvement in the fatigue strength of stainless steel by the cavitating jet in air was also demonstrated compared with that of a cavitating jet in water [28].

To treat a large area, a nozzle consisting of multiple orifices was proposed, and the improvement in fatigue strength of stainless steel was reported [66]. To treat both sides at one time, cavitation peening using opposed cavitating jets was proposed and the fatigue strength of a duralumin plate with a hole was improved [67].

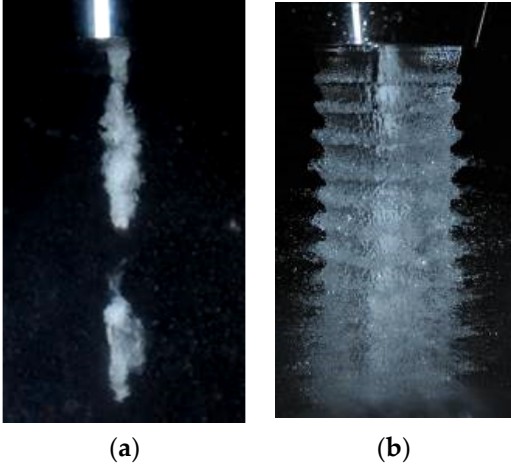

(**a**)    (**b**)

**Figure 9.** Aspect of cavitating jet in water and in air [54]: (**a**) cavitating jet in water and (**b**) cavitating jet in air.

### 3.2.2. Ultrasonic Cavitation

Cavitation erosion resistance of materials was evaluated using a vibratory test, in which cavitation is generated by ultrasonic vibration [6]. Note that a material test using hydraulic cavitation was standardized by ASTM International [7]. Cavitation induced by ultrasonic vibration is called ultrasonic cavitation. One of major topic related to ultrasonic cavitation is sonochemistry, in which chemical reactions are accelerated by ultrasonic cavitation [68]. Even hydraulic cavitation can be used for chemical processes, and the efficiency of pretreatment of biomass using hydraulic cavitation was found to be 20 times better than that of ultrasonic cavitation [69].

Figure 10 shows typical aspect of ultrasonic cavitation [70]; cavitation was observed at the tip of a vibratory horn. In Figure 10, the aspect of cavitation was observed by an instantaneous photography using a flush lamp. As the aspect of cavitation induced by ultrasonic vibration changed with time, the flush lump was synchronized with the vibration. In Figure 10, the white bubbles are cavitation. At bubble collapse, a shockwave was induced and propagated, then the black circle region, i.e., shock ring, shows where a shockwave was propagated with collapsing the bubble. Compressive residual stress was introduced into stainless steel powders by ultrasonic cavitation [26], and a machining process used created using ultrasonic cavitation [71]. Compressive residual stress was also introduced into metallic materials [72,73]. As shown in Figure 10, cavitation occurs on the tip of the vibratory horn. The effect of horn-tip geometry on ultrasonic cavitation peening was investigated [74]. During ultrasonic cavitation peening, the vibratory horn is placed to the target surface at a certain distance from the target. The aggressive intensity of ultrasonic cavitation drastically changes with the distance from the target [70], which was confirmed both experimentally and theoretically [75]. Unfortunately, the sensitivity of aggressive intensity to the distance using ultrasonic cavitation limits the practical applications. Internal surface finishing was proposed using ultrasonic and abrasive cavitation [76], and the internal surface finishing research has shifted to hydrodynamic abrasive cavitation [77]. Ultrasonic cavitation peening was also used to improve micro-burr-free surfaces [78].

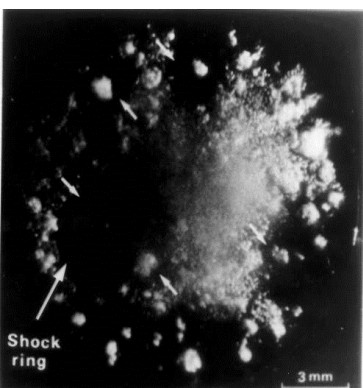

**Figure 10.** Aspect of cavitation induced by vibratory apparatus [70]. Reproduced with permission from The Japan Society of Mechanical Engineers.

### 3.2.3. Laser Cavitation

The other typical method used to create cavitation bubbles is a pulse laser [52]. With laser peening, the target can be treated in two ways: a pulse laser is exposed to the target with a water film [79–84], or the pulse laser irradiates the target placed in water [3,85–87], which is called submerged laser peening. For both laser peening methods, the shockwave induced by laser ablation causes plastic deformation on the material due to containment by the inertia of water. With submerged laser peening, a bubble is generated after laser ablation in the same way to create laser cavitation, as mentioned above. The amplitude of the pressure wave caused by the laser ablation was found to be larger than that of laser cavitation when the shockwave in water was measured using a submerged shockwave

sensor [88]. However, the impact induced by the collapse of laser cavitation is larger than that of laser ablation [3,86] when the impact force passing through in the target is measured by a polyvinylidene fluoride (PVDF) sensor, which was developed to detect cavitation impact energy [89,90]. Figure 11 shows typical laser ablation (LA) and laser cavitation (LC) with the output signal from a PVDF sensor and a submerged shockwave sensor. The details are provided in [3,86]. LA and LC were observed using a high-speed video camera. The used pulse laser was a Q-switched Nd:YAG laser with 1064 nm wavelength, a maximum energy of 0.35 J, and a pulse width of 6 ns. The handmade PVDF sensor was installed in the target and the submerged shockwave sensor was placed in water near the target. As mentioned above, the amplitude of the signal from the PVDF sensor at LC collapse was larger than that of LA. The amplitude of the signal from the submerged shockwave sensor with LA was larger than that of LC. Thus, when the focus point of the pulse laser was set in the water, LC can peen the surface without LA [91]. As shown in Figure 11b, the impact induced by secondary collapse was detected at $t = 1.54$ ms by the PVDF sensor, although the secondary collapse could not be observed by the submerged shockwave sensor (Figure 11c). This is one of reasons why the measurement using the PVDF sensor is reasonable. When the pressure was measured by a hydrophone, the amplitudes of LA and LC were nearly equivalent [92].

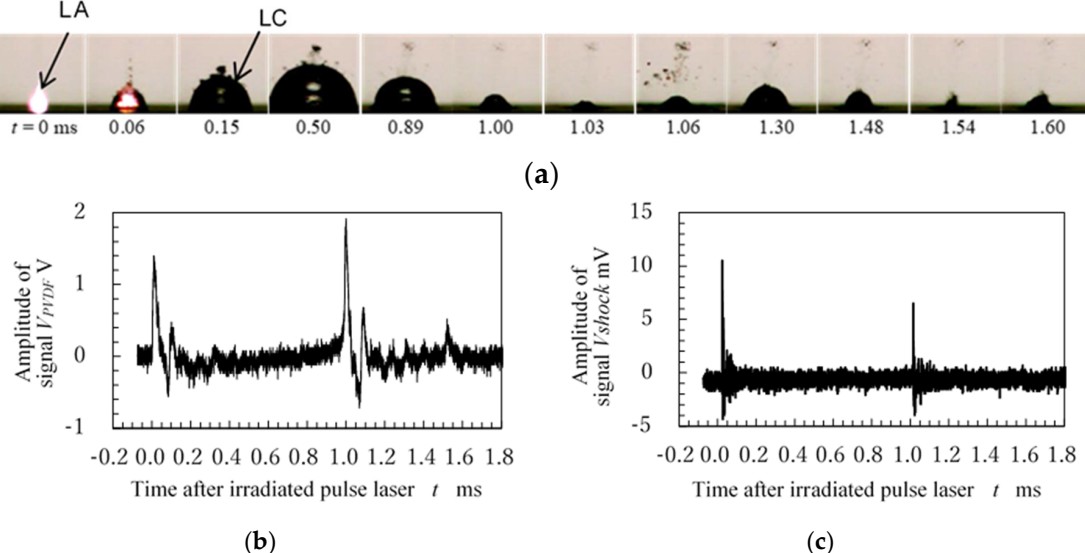

**Figure 11.** Laser ablation (LA) and laser cavitation (LC) produced by a pulse laser. (**a**) Aspect of laser ablation (LA) and laser cavitation (LC); (**b**) Signal from polyvinylidene fluoride (PVDF) sensor; (**c**) Signal from submerged shockwave sensor.

## 4. Key Parameters of Cavitation Peening

### 4.1. Type of Cavitating Jet

With the conventional method to create cavitation, a submerged high-speed water jet with cavitation, i.e., a cavitating jet in water, is used. As mentioned above, to treat the outside surface of tanks, pipelines, and other components, a cavitating jet in air was constructed [27,64] and the fatigue strength was improved using the cavitating jet in air [28]. A cavitating jet in air is used for the treatment of a skin pass mill work roll [93]. To treat valuable components, such as biomedical implants and/or dies, a cavitating jet in water with a pressurized chamber is used [38,94]. To clearly differentiate the type of cavitating jet on the peening effect, Figure 12 depicts introduced compressive residual stress into stainless steel SUS316L by cavitation peening (CP) using a cavitating jet in water, a cavitating jet in air, and a cavitating jet in a water-pressurized chamber [3]. The surface compressive residual stress introduced by a cavitating jet in air was shown to be larger than that of a cavitating jet in water [3,95]. Note that a cavitating jet in water introduces compressive residual stress in a deeper region than a

cavitating jet in air [3,95]. The compressive residual stress introduced using a cavitating jet in water with a pressurized chamber is larger and deeper compared with the others, as shown in Figure 12 [3].

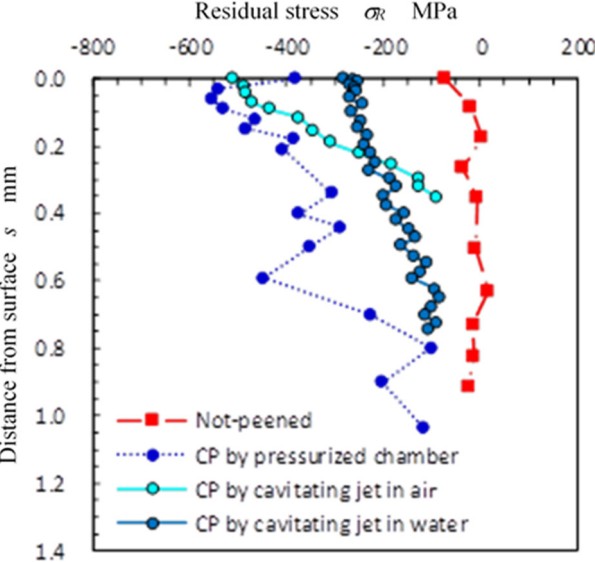

**Figure 12.** Compressive residual stress introduced by several types of cavitating jets (stainless steel) [3].

## 4.2. Standoff Distance

One of most important parameters of cavitation peening is standoff distance, which is defined as the distance from the nozzle to the target [96]. The standoff distance is precisely defined by the distance from the upstream corner of the nozzle to the target, as the flow separates at the corner. Figure 13 shows the relationship between the standoff distance and the curvature $1/\rho$, which is calculated from the arc height and chord length of the peened specimen, as shown in the Appendix A, at the injection pressures $p_1$ of 40 and 60 MPa [97]. For shot peening, an Almen strip, whose width is 19 mm, is treated, and the arc height with a chord length of 40 mm is measured, then the arc height is used to evaluate peening intensity. As shown in the Appendix A, the arc height changes with the chord length; however, the curvature is not affected by the chord length. Thus, the curvature $1/\rho$ is used in the present review. The $1/\rho$ has two peaks, i.e., the first peak on the near side of nozzle and a second peak on the far side. The first peak is caused by water jet peening and the second by cavitation peening. As shown in Figure 6a, as the cavitation develops and then collapses on the target, a certain distance from the nozzle is required. When the target is set too close to the nozzle, the target is impinged by water columns in the jet center, i.e., water jet peening. As shown in Figure 13, the $1/\rho$ of the second peak at $p_1 = 40$ MPa is larger than that of the first peak of $p_1 = 60$ MPa. The peening intensity of optimized cavitation peening is larger than that of water jet peening. The details of the differences between water jet peening and cavitation peening are explained in Section 6. Lichtarowicz identified the relationship between the optimum standoff distance $s_{opt}$, where the aggressive intensity of a cavitating jet has a peak and cavitation number $\sigma$, as expressed in Equation (3) [98]:

$$\frac{s_{opt}}{d} \propto \sigma^{-n} \tag{3}$$

where $d$ is the nozzle throat diameter and $n$ is a constant. Note that $n$ depends on the nozzle geometry [39]. The optimum standoff distance of similar nozzle geometry can be estimated using Equation (3) at various injection pressures.

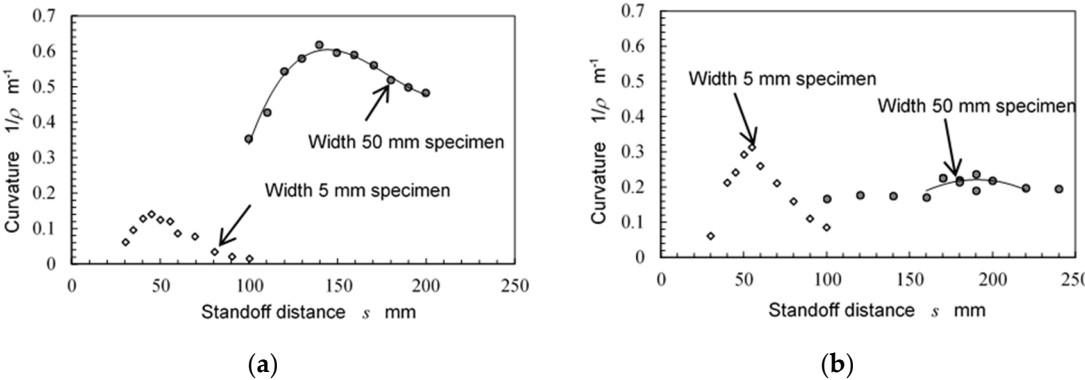

**Figure 13.** Peening intensity as a function of standoff distance [97]: (**a**) 40 and (**b**) 60 MPa. Reproduced with permission from The Water Jet Technology Society of Japan.

### 4.3. Nozzle Geometry and Diameter

The aggressive intensity of a cavitating jet through a conventional water jet nozzle is very low, although water jets are used for generating cavitation. Figure 14 illustrates a schematic diagram of the relative aggressive intensity of jets from data published in previous reports [39,41]. In Figure 14, nozzles A–E are conventional nozzles for a water jet. Nozzle F is a standard nozzle for material testing using a cavitating jet [7]. As shown in Figures 5 and 6, the vortical flow near nozzle is very important for generating a powerful cavitating jet. Nozzle G is an optimized nozzle with vortical flow in the outlet geometry at the nozzle exit [40]. As cavitation is initiated from cavitation nuclei, a cavitator that generates the nuclei was set upstream of the main nozzle (nozzle H); the aggressive intensity was nearly two times larger than without the cavitator (nozzle G) [41]. When the guide pipe, which enhances the vortical flow and the cloud cavitation, was placed downstream of the nozzle (nozzle I), the aggressive intensity was nearly two times that of nozzle G. When both the cavitator and the guide pipe were used, the aggressive intensity was nearly four times larger than that of nozzle G. The effect of nozzle geometry was also previously discussed [99]. To enhance the flow rate of the cavitating jet at a constant water jet flow rate, water flow holes near the nozzle outlet were used [43]. Note that the nozzle geometries of nozzles F–J are simple and the erosion in the nozzle is minimal compared with the horn-type nozzle.

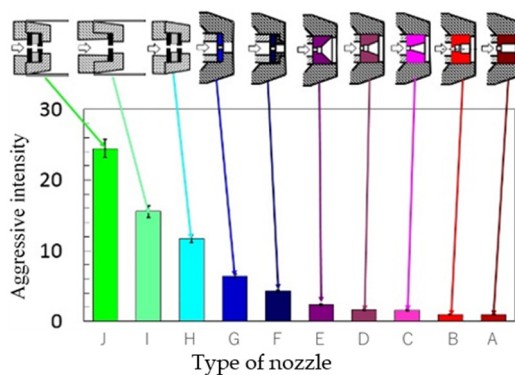

**Figure 14.** Schematic diagram of aggressive intensity of the cavitating jet using several types of nozzles.

The other important parameter related to the nozzle is nozzle size, i.e., diameter of the nozzle throat. To demonstrate the peening effect of nozzle size, Figure 15 shows residual stress as a function of the depth from the surface with changing nozzle diameter from 0.35 to 2 mm at $p_1 = 30$ MPa and $p_2 = 0.1$ MPa [100]. The tested material was stainless steel SUS316L. As shown in Figure 15, a larger nozzle can introduce larger compressive residual stress into a deeper region of the stainless steel. The

effect of nozzle diameter was also investigated from 0.4 to 0.6 mm at $p_1$ = 16.5 and 16.7 MPa [99]. The scale effect of nozzle size on the aggressive intensity of a cavitating jet was experimentally evaluated, and the power law was reported: the exponent depends on cavitation number [101]. The exponent is 1.56 ± 0.03 at $\sigma$ = 0.01, 1.97 ± 0.03 at $\sigma$ = 0.014, and 2.49 ± 0.02 at $\sigma$ = 0.02 [101].

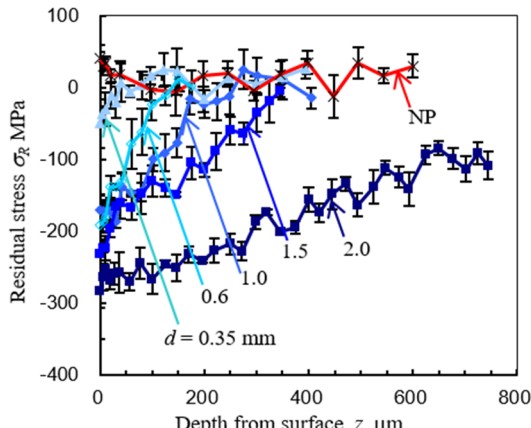

**Figure 15.** Effect of nozzle diameter on introduction of compressive residual stress compared with non-peened (NP) (stainless steel) [100].

### 4.4. Injection Pressure

To reveal the effect of injection pressure at a constant downstream pressure on the introduction of compressive residual stress, Figure 16 shows the residual stress as a function of the depth from the surface. Note that the cavitation number changes with the injection pressure at a constant downstream pressure, as shown in Equation (2). The tested material was the same stainless steel as in Figure 15. The diameter of the nozzle was 0.35 mm and $p_2$ was 0.1 MPa. The effect of cavitation number is explained in the next section. As shown in Figure 16, the compressive residual stress and the depth of the compressive layer barely increased with injection pressure. Comparing the residual stress distribution of $d$ = 2 mm at $p_1$ = 30 MPa in Figure 15 with that of $d$ = 0.35 mm at $p_1$ = 300 MPa in Figure 16, a cavitating jet using a large nozzle at relatively low injection pressure introduces larger compressive residual stress into a deeper region. Note that the jet power of $d$ = 0.35 mm at $p_1$ = 300 MPa and that of $d$ = 2 mm at $p_1$ = 30 MPa are nearly equivalent. A plunger pump of 300 MPa is expensive compared with a 30 MPa pump. When larger nozzles, such as 2 mm in diameter, are used, 10 MPa would be enough to introduce compressive residual stress into stainless steel [102].

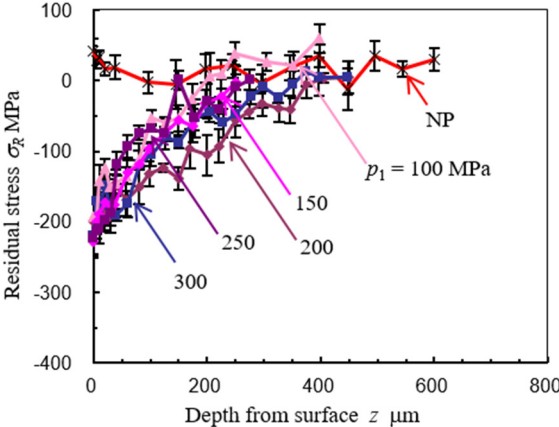

**Figure 16.** Effect of injection pressure on introduction of compressive residual stress (stainless steel) [100].

### 4.5. Cavitation Number

As cavitating flow is used for cavitation peening, cavitation number $\sigma$, which is defined by Equation (2), is one of the most important parameters. The cavitating region increases with a decrease of cavitation number [2,47]. The aggressive intensity of cavitation through a Venturi tube does not monotonously increase with cavitating length, i.e., a decrease of cavitation number; it peaks at a certain cavitation number [47]. Figure 17 shows the normalized aggressive intensity of a cavitating jet as a function of cavitation number at constant injection pressure [3]. At constant injection pressure, cavitation number increases with increasing downstream pressure. At higher downstream pressures, cavitation bubbles collapse violently, although the bubbles become smaller. At too low a cavitation number, i.e., too low downstream pressure or too high injection pressure, the bubble collapse becomes soft. When cavitation number decreases, the cavitating region increases, but the individual impacts strengthen. This is why the aggressive intensity of the jet peaked at $\sigma = 0.01$–$0.02$, as shown in Figure 17. The aggressive intensity of a cavitating jet with $p_1 = 98$ MPa had a peak at $\sigma = 0.01$–$0.014$ [103], and the jet with $p_1 = 20$ MPa had a peak at $\sigma = 0.014$ [104]. The cavitation number, where it peaked, slightly changed with nozzle outlet geometry, but was in the range of 0.01 to 0.02 [105]. As the downstream pressure at the conventional cavitation peening using an open water-filled chamber is at nearly atmospheric pressure, i.e., $p_2 \approx 0.1$ MPa, from the cavitation number viewpoint, the aggressive intensity of the jet decreases with increasing injection pressure at $p_1 > 10$ MPa as $\sigma < 0.01$. From the injection pressure viewpoint, the aggressive intensity of the jet increases. Thus, total aggressive intensity of the jet peaks as a function of injection pressure. The details are provided in Figure 19 as referred to hereinafter.

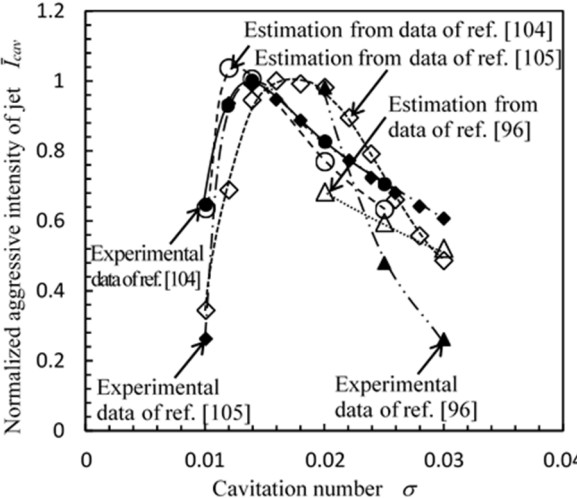

**Figure 17.** Normalized aggressive intensity of a cavitating jet as a function of cavitation number [3].

### 4.6. Geotrty Effect of Treatment Section

At conventional cavitation peening, a high-speed water jet is injected into a water-filled chamber. As cavitation is hydrodynamic phenomenon, flow pattern affects aggressive intensity of the jet which relates the peening intensity. As it was reported that the erosion induced by the cavitating jet was affected by the size and the geometry of the chamber [106,107], the peening intensity changes with the size and geometry of the chamber.

The peening effect of the water depth of an open chamber was investigated, and the authors found that the water depth needs to be considered to mitigate the cushioning effect caused by air bubbles entrained into the jet [3]. The other main factors are the attack angle of the jet and the shape of the target. The effect of incident angle of the cavitating jet was reported [99,108]. When convex and concave surface were treated by a cavitating jet, the peening area changed with the curvature of the surface [109].

### 4.7. Scanning Pitch and Speed

As shown in Figure 8, the peening area of the fixed cavitating jet is a ring pattern. To treat a wider area, the jet should scan the target or the target should be moved with a fixed jet. The optimum overlapping scanning pitch of the jet was revealed by estimating the aggressive intensity distribution of the jet [110].

As the individual impact produced by cavitation bubble collapse causes individual plastic deformation, and the compressive residual stresses result from total summation of the plastic deformation, the introduced residual stress $\sigma_R$ changes with processing time per unit length $t_p$ as described by Equation (4) [65]:

$$\sigma_R = (\sigma_{sat} - \sigma_0)\left(1 - e^{-a\,t_p}\right) + \sigma_0 \tag{4}$$

where $\sigma_0$ and $\sigma_{sat}$ are initial and saturated residual stress, respectively, and $a$ is a constant.

### 4.8. Water Qualities

As cavitation is a phase change phenomenon from liquid to gas, the temperature and gas content affect the aggressive intensity of the cavitation bubble collapse. The effect of temperature and gas contest on cavitation erosion was revealed using a vibratory cavitation erosion test [111–113]. For a cavitating jet, the effect of temperature on erosion rate was confirmed [114–116]. The peening intensity was measured as 278–308 K, and the peening intensity was nearly constant at 288–308 K [3]. Han et al. tried to enhance cavitation peening intensity by aeration; however, they concluded that further investigations and process optimization still need to be conducted [35]. As mentioned in Section 4.3, although cavitation nuclei are required for the cavitating jet, the presence of too many air bubbles reduces the impact due to the cushion effect [2]. The usage of oil for cavitation peening was proposed, and compressive residual stress into aluminum alloy was introduced [117].

### 4.9. Material Properties

To estimate or avoid cavitation erosion, the relationship between cavitation impact and material properties such as hardness was investigated [118]. As cavitation erosion is a kind of fatigue failure, a concept of fundamental threshold level on cavitation erosion was suggested [119]. An experimental method used to obtain the threshold level of materials using a cavitating jet apparatus was proposed, and the threshold levels of metals and plastics were identified [8]. In the case of a single cavitation bubble induced by a pulse laser, the threshold level on peening effect was revealed experimentally, which was confirmed by numerical simulation [120]. To treat harder materials, severe impacts are required or a longer treatment time.

## 5. Estimation of Aggressive Intensity of Cavitation Peening

As mentioned in Section 4.5, when injection pressure increases, cavitation number decreases. From the cavitation number viewpoint, in the region of $\sigma < 0.01$, i.e., $p_1 > 10$ MPa at $p_2 = 0.1$ MPa, increases in the injection pressure decrease the aggressive intensity of the jet. The increase in the injection pressure can increase the aggressive intensity of the jet at a constant cavitation number. Thus, to estimate the aggressive intensity of the jet $I_{cav\ est}$ at the optimum standoff distance from injection pressure $p_1$ and nozzle diameter $d$, an experimental formula was proposed as follows [101]:

$$I_{cav\ est} = I_{cav\ ref}\ K_n\ \frac{f(\sigma)}{f(\sigma_{ref})}\left(\frac{d}{d_{ref}}\right)^{n_d}\left(\frac{p_1}{p_{1\ ref}}\right)^{n_p} \tag{5}$$

where $I_{cav\ ref}$ is a reference aggressive intensity of a jet under the reference condition; the parameters with the subscript $_{ref}$ are those for the reference conditions, $K_n$ depends on the geometry of the nozzle

and/or the test section, and $n_d$ and $n_p$ are the exponents of the power laws. The *f (σ) is* a function of cavitation intensity and is related to cavitation number, and the following equation was proposed [3]:

$$f(\sigma) = \sigma^{-1.8} \left\{ p_1(\sigma - \sigma_s) - p'_v \right\} \tag{6}$$

where $\sigma_s$ and $p_v'$ are the correlation factors of the cavitation number and the pseudo vapor pressure, respectively. The aggressive intensity of the jet was estimated using Equation (5) and the difference between the estimated and experimental values is 16% at constant $\sigma$ [3].

## 6. Difference between Cavitation Peening and Water Jet Peening

### 6.1. Standoff Distance

The term 'water jet peening' has been used for different peening methods, as shown in Table 1 [95]. To avoid misunderstanding of peening mechanism, the term 'cavitation peening' is used for the peening method using cavitation impact, and water jet peening is used for the peening method using water column impacts.

As mentioned in Section 4.2, the first peak results from water jet peening, the second peak is caused by cavitation peening, and the optimum standoff distance $s_{opt}$ is shown in Equation (3) [98]. Then, a classification map for cavitation peening and water jet peening was proposed considering cavitation number $\sigma$ and $s_{opt}$, which was normalized by the nozzle throat diameter $d$, as shown in Figure 18 [95]. More than 150 points [17,39–41,96,100,102,106,121–127] are plotted in Figure 18, and the line described by Equation (7) divides two regions, i.e., the first and second peak regions. The upper right hand region is cavitation peening and the lower left hand side region is the water jet peening region.

$$\frac{s_{opt}}{d} = 1.8 \, \sigma^{-0.6} \tag{7}$$

The usage of Figure 18 is as follows:

(1) Find the optimum standoff distance $s_{opt}$ changes with distance from the nozzle to the target using experiments.
(2) Calculate $s_{opt}/d$.
(3) Calculate $1.8 \, \sigma^{-0.6}$ using Equation (2). Note that $p_1$ and $p_2$ in Equation (2) are absolute pressures.
(4) If experimental $s_{opt}/d$ is larger than the $s_{opt}/d$ obtained in step (3), the peening is cavitation peening.
(5) If experimental $s_{opt}/d$ is smaller than the $s_{opt}/d$ obtained in step (3), the peening is water jet peening. If cavitation peening is desired, find the other optimum standoff distance on the far side of the nozzle using experiments.

**Table 1.** Colloquial terms for water jet peening [95].

| Method | Mechanism of Impact | Source of Impact |
| --- | --- | --- |
| Wet shot peening | Solid collision | Shot |
| Water jet peening | Liquid collision | Water droplet |
| Cavitation peening | Shockwave | Cavitation collapse |

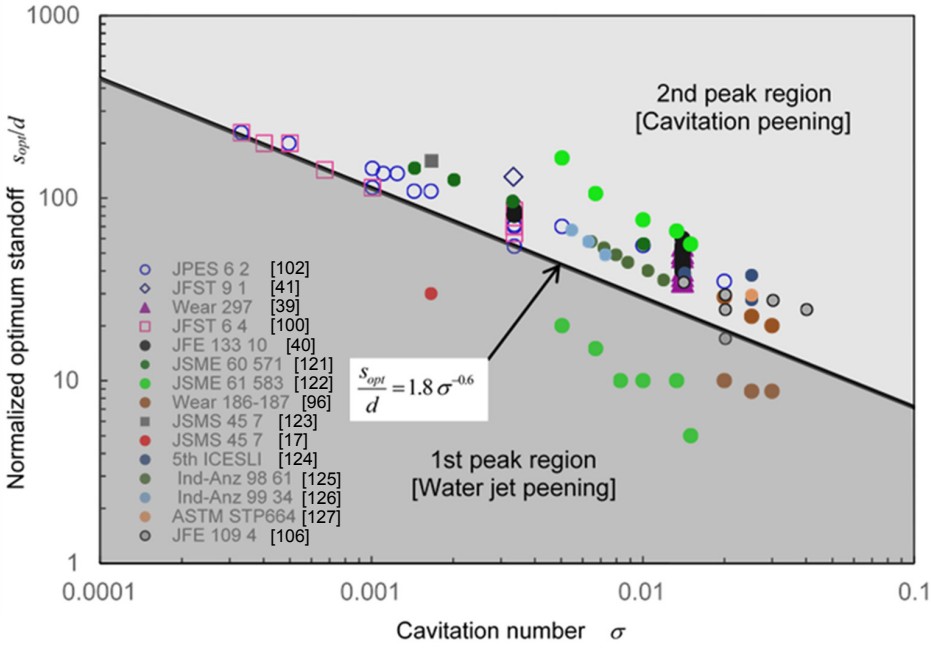

**Figure 18.** Classification map for cavitation peening and water jet peening. [95]

*6.2. Peening Intensity*

To compare the peening intensity between water jet peening and cavitation peening, Figure 19 depicts the peening capability $\beta$ calculated from arc height changes with injection pressure at constant downstream pressure considering treatment width [97]. For both water jet peening and cavitation peening, the optimum standoff distance was obtained by measuring the arc height with the standoff distance, then the peening capability at the optimum standoff distance was evaluated. As shown in Figure 19a, for water jet peening, $\beta$ increases with injection pressure. When the power law is assumed, the exponent is about 2.2. In the case of cavitation peening, $\beta$ has a maximum at $p_1 = 40$ MPa. Note that the maximum value of cavitation peening is 1.7 times larger than that of water jet peening. As the jet power, which is defined by the injection pressure and the flow rate, of 60 MPa is 1.8 times larger than that of 40 MPa, the peening efficiency of cavitation peening is about three times better than that of water jet peening.

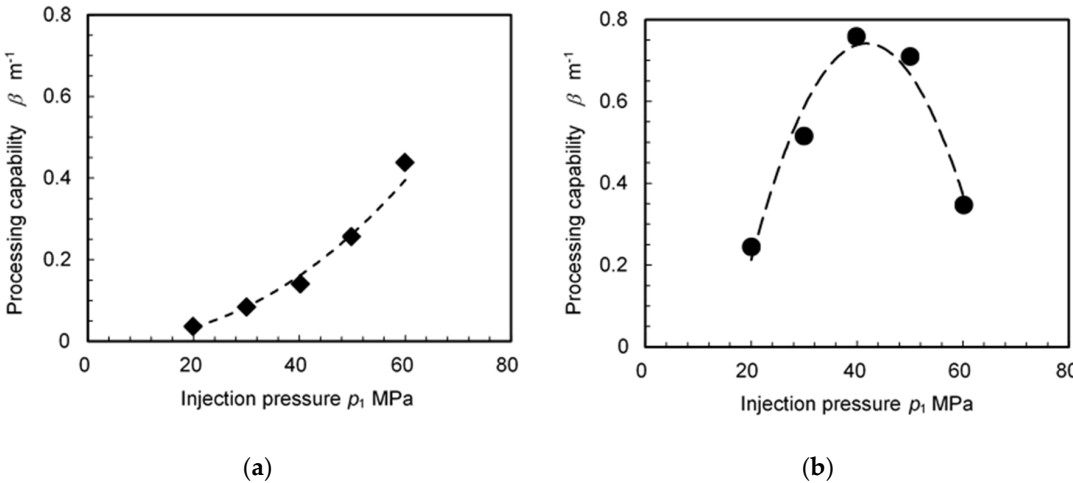

(**a**)　　　　　　　　　　　　　(**b**)

**Figure 19.** Processing capability as a function of injection pressure [97]. Reproduced with permission from The Water Jet Technology Society of Japan; (**a**) Water jet peening; (**b**) Cavitation peening.

## 7. Comparison between Cavitation Peening and Other Peening Methods

To compare cavitation peening (CP) with the other peening methods, such as shot peening (SP), laser peening (LP), and water jet peening (WJP), Figure 20 shows the results of a plane bending fatigue test of stainless steel [54]. In Figure 20, the amplitude of bending stress was normalized by the fatigue strength of non-peened amplitude of 279 MPa. For all cases, the optimum processing time or optimum pulse density was obtained from the changes in fatigue life at 400 MPa with processing time or pulse density [86]. As shown in Figure 20, the fatigue life of shot peening at 400 MPa was longer than that of cavitation peening; however, the fatigue strength of cavitation peening was stronger than that of shot peening.

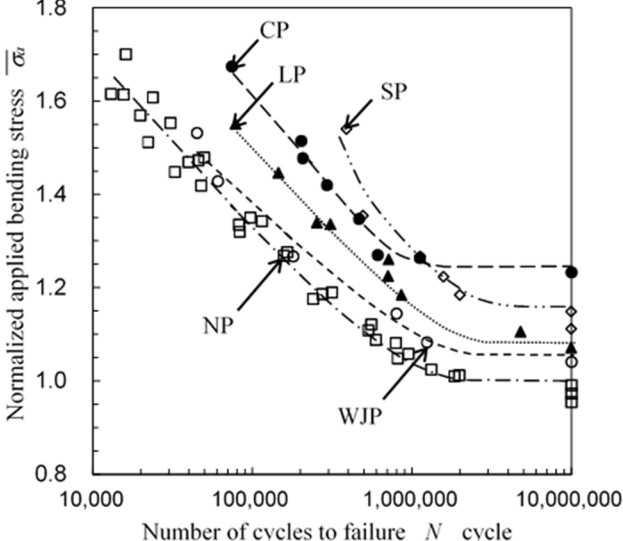

**Figure 20.** Improvement in fatigue strength of stainless steel by cavitation peening (CP), shot peening (SP), laser peening (LP), and water jet peening (WJP) compared with non-peened (NP).

A singular characteristic of cavitation peening on the microstrain of quenched tool steel alloy is that the microstrain was relieved by introducing compressive residual stress [128,129]. As expected, many dislocations were introduced by quenching and/or machining. These dislocations can be moved to the grain boundaries and disappear using high frequency vibrations induced by cavitation impacts, as the cyclic loading or ultrasonic vibration move dislocations [130–132]. In the case of conventional shot peening, dislocations increased due to solid collision. However, for cavitation peening, the disappearance of dislocations is more frequent compared with the increase in the dislocations, as no solid collisions occur with cavitation peening. The possibility of the movement of dislocations was confirmed by observation using a transmission electron microscope (TEM) [133].

The microstructures of metallic materials treated by cavitation peening were investigated by comparison with other processes [134–136]. The increase in surface roughness was less than that of the other processes [134]. When the plastic deformation inside the material was evaluated using the Fry etching method, the depth of the dent of cavitation peening and laser peening was shallower than that of shot peening [137,138], resulting from the difference in the strain speed of the process, as the plastic deformation of cavitation peening and laser peening is caused by the shockwave process.

## 8. Application of Cavitation Peening

### 8.1. Suppression of Environmental Assisted Cracking

As mentioned in Section 2, at the beginning, the main purpose of cavitation peening is mitigation of stress corrosion cracking [10,123], which was successfully applied to nuclear power plants [11].

Although cavitation causes erosion and corrosion, cavitation improved the corrosion resistance, as cavitation can generate inactivity layer by oxidation [139]. The electrochemical characteristics induced by cavitation peening were also reported [140].

From the environmental-assisted cracking viewpoint, hydrogen embrittlement and delayed fracture are similar to stress corrosion cracking. The suppression of hydrogen-assisted fatigue crack growth in austenitic stainless steel and delayed fracture resistance on chrome molybdenum steel by cavitation peening were reported [141,142].

*8.2. Improvement of Fatigue Properites*

Improvements in the fatigue strength of metallic materials, such as aluminum alloy [4,117,143], carbonized steel [12], nitrocarburized steel [29], stainless steel [28], silicon manganese steel [32], and titanium alloy [38], by cavitation peening were reported. Additive manufactured titanium alloy was also tested in comparison with shot peening and laser peening, and the fatigue strength was improved by cavitation peening by about two times in comparison with non-peened [44]. The reasons for the improvement are the increase in yield stress by work hardening and the introduction of compressive residual stress [45]. The actual mechanical components, such as CVT elements [13] and gear [14,15], were treated by cavitation peening and their improvements in fatigue strength were reported. Figure 21 shows a cavitating jet injecting to the gear. The geometry of the gear in Figure 21 is that of the gear tested in the reference [15]. To clarify the mechanism of improvement in fatigue strength by cavitation peening, the crack initiation and crack growth of a surface modified layer treated by cavitation peening were evaluated in comparison with other peening methods [144,145].

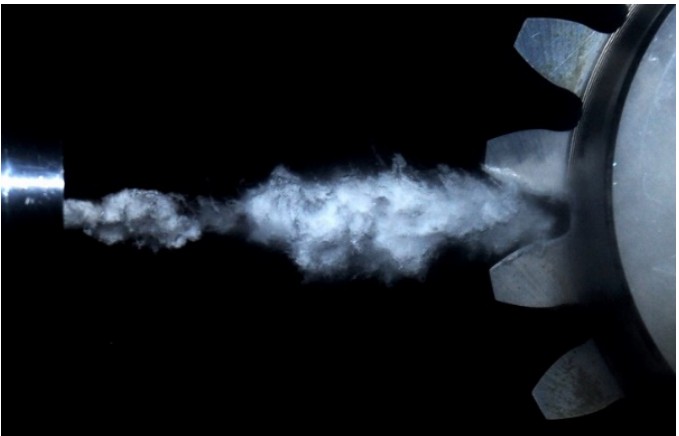

**Figure 21.** Aspect of cavitating jet injecting to the gear.

*8.3. Improvement in Triborogical Properties*

As cavitation peening is shotless peening, as mentioned above, cavitation peening can introduce compressive residual stress and work hardening with less of an increase in surface roughness in comparison with other processes. Thus, cavitation peening improves the fretting fatigue properties and/or pitting resistance [37,146–148]. The pressure distribution of individual cavitation impact was evaluated using a magnesium-oxide single crystal, and the authors concluded that the highly intense impact was applied to the center [149]. Figure 22 shows the typical plastic deformation pit induced by cavitation impact on pure aluminum compared with ball indentation, which is a model of shot impact [150]. As shown in Figure 22, the edge of the plastic deformation pit induced by cavitation impact is smoother compared with that of ball indentation. The pit induced by the cavitation impact seems to be suitable for an oil pool. This shape can be used for tribological applications.

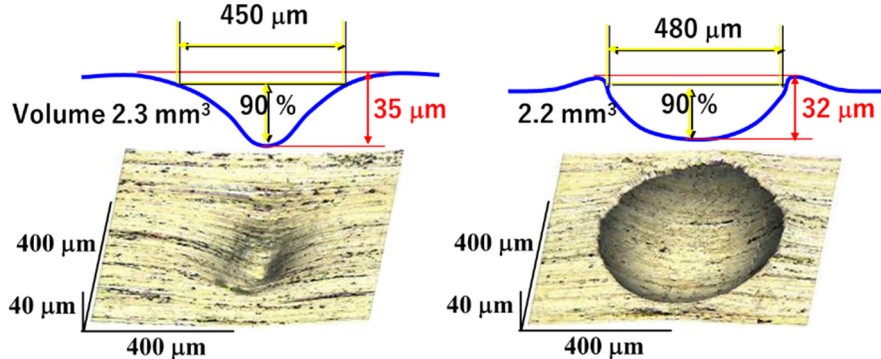

**Figure 22.** Typical aspect of plastic deformation pit induced by cavitation impact in comparison with ball indentation. (**a**) Cavitation impact; (**b**) Ball indentation.

*8.4. Gettering Effect of Silicon Wafer*

Cavitation peening can be used for silicon wafers as well as metallic materials, as cavitation impact can be used for the gettering technique. The gettering technique is an essential method used to remove unwanted impurities from active device regions in the manufacturing of semiconductors [151–155]. The most popular and conventional gettering technique is the introduction of backside damage of silicon wafer by impact of $SiO_2$ particles [156,157]. However, the fractions of the particle form an additional source of contamination during subsequent wafer processing. Cavitation peening can introduce backside damage without contamination, as shots are not required. Kumano et al. successfully applied cavitation impacts for the gettering technique, proved by photo-capacitance measurements [158]. Note that impacts that introduce strain without cracks are required for the introduction of the gettering site. This is a typical demonstration that the cavitation impacts induced by the cavitating jet are controllable.

## 9. Conclusions

In the present review, to reliably and safely apply cavitation peening in practical applications, the research and applications of cavitation peening were reviewed in comparison with water jet peening and other peening methods. To use cavitation peening, the most important aspect is experimental classification between cavitation peening and water jet peening. For cavitation peening, a submerged water jet with a large nozzle at low injection pressure is suitable. The important characteristics of cavitation peening are summarized as follows:

(1)  Cavitation peening can mitigate environmentally assisted cracking, such as stress corrosion cracking, hydrogen belittlement, and delayed fracture. It is also applied for the improvement of fatigue strength and tribological properties of metallic materials.

(2)  For conventional cavitation peening, a submerged high-speed water jet is used. However, the peening mechanisms of cavitation peening and water jet peening are different. Cavitation peening uses the impacts of bubble collapses. Water jet peening uses water column impacts. Note that submerged high-speed water can also treat the target by water jet peening.

(3)  For cavitation peening using an open water-filled chamber, peening intensity has a maximum at a certain injection pressure of about 40 MPa. The peening intensity of cavitation peening at 40 MPa is larger than that of water jet peening at 60 MPa.

(4)  Cavitation peening and water jet peening are classified by the relationship between cavitation number and the normalized distance from the nozzle to the target.

(5)  Submerged laser peening is a kind of cavitation peening using the impact at the collapse of the bubble, which develops after the laser ablation when it is optimized.

**Author Contributions:** H.S. conceived and designed the paper; H.S. performed the additional experiments and wrote the paper.

**Funding:** This research was partly supported by JSPS KAKENHI grant numbers 17H03138 and 18KK0103.

**Conflicts of Interest:** The author declares no conflict of interest.

## Appendix A

Figure A1 depicts a schematic diagram of curvature $1/\rho$ obtained from the arc height $h_1$ of the chord length $L_1$. Using Pythagorean theorem, radius of curvature $\rho$ is expressed by $h_1$ and $L_1$ as shown in Equation (A1). As shown in Equation (A2), radius of curvature $\rho$ is derived from Equation (A1). Then, the curvature $1/\rho$ is obtained, as shown in Equation (A3), which is proportional to the arc height, which is normally used for Almen strips to evaluate the intensity of shot peening. As shown in Figure A1, when a different chord length $L_2$ is used, the arc height $h_2$ is different. However, curvature $1/\rho$ is a unique parameter. Thus, the curvature $1/\rho$ is used in the present review instead of the arc height.

$$\rho^2 = (\rho - h_1)^2 + \left(\frac{L_1}{2}\right)^2 \tag{A1}$$

$$\rho = \frac{L_1{}^2}{8h_1} + \frac{h_1}{2} \cong \frac{L_1{}^2}{8h_1} \tag{A2}$$

$$\frac{1}{\rho} \cong \frac{8h_1}{L_1{}^2} \tag{A3}$$

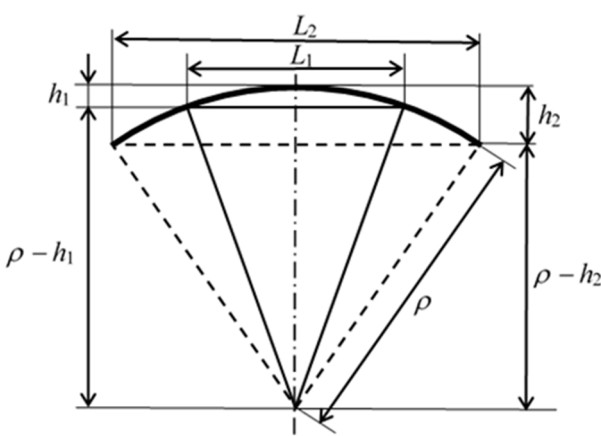

**Figure A1.** Schematic diagram of radius of curvature and arc height.

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
