# Peer review of "Cavitation Peening: A Review"

_metals, doi:10.3390/met10020270_

Round 1

Reviewer 1 Report

The paper is well organized and fills a gap in scietific literature of peening treatments. Minor English check are needed before it can be published.

Author Response

Thank you for reviewing my manuscript. My manuscript was proofread and corrected by MDPI’s English editing service.

Reviewer 2 Report

I am more familiar with shot peening and laser peening and accepted this review to improve my awareness of cavitation peening. As a review of a less well-known subject I think it is potentially a very useful contribution to the literature.

However, there were several issues with this paper that need attention.

Regrettably, the English is poor and this makes understanding the technical description difficult, especially for a non-expert. 

The technical descriptions too often assume a familiarity with cavitation peening and, added to the poor English, make understanding difficult. More elaboration for the non-expert would be useful. For example (but there are many), the section on Standoff Distance is not clear, with lack of explanation  of curvature etc. that would be accessible  to non-expert.

The figures were not always well explained. Examples are Figures 4, 10, 11.  The last is particularly poor.

There are also a number of  typographical errors and the figure numbers should be checked.

Reviewer 3 Report

Review for paper entitled: “Cavitation peening: A review”

Author: Hitoshi Soyama

General comment: In this paper author present a great revision on the topic of cavitation peening, which is used in surface modification technology to enhance the mechanical properties of treated component.  He provided a brief history of the cavitation peening since the early stage of the development to the current status. He also expressed the definition of the cavitation as well as generation mechanism of hydraulic cavitation, ultrasonic cavitation, and laser cavitation. Author also analyze the main parameters of cavitation peening such as cavitating jet, stand-off distance, nozzle geometry and inner diameter, inject pressure, cavitation number, geometry effect of treatment section, scanning pitch and speed, water quality, and material properties. An empirical formulation of cavitation peening intensity links to injection pressure, cavitation number, stand-off distance, nozzle diameter, etc. Furthermore, author also classified the cavitation peening and jet peening from some different point of view such as stand-off distance used, peening intensity magnitude, and effective inject pressure. In addition, author did some comparisons among different peening techniques to highlight the benefits of cavitation peening in term of fatigue strength, life cycle of treated components. Finally, author summarized the applications of the cavitation peening.

It is great paper, reviewer recommends to publish in this journal. However, author must address the following concerns.      

1)      Since both water jet peening and cavitation peening occur in the same process as water jet peening mainly concentrates at the center point, while the cavitation mainly at the ring area. Thus, it would be great if author can provide or elaborate more on the cavitation peening mechanism, such as bubble formation, bubble trajectory, bubble collapse, shock wave formation, and impaction on the substrate, strength of shock or pressure at the impact area.

2)      Since experiment may not capture or might not provide a high resolution to provide a details of the cavitation peening phenomena. It is necessary to provide more numerical works or analytical works to have full view of the cavitation peening mechanism.

3)      It seems that cavitation peening in air is more efficiency comparing to cavitation peening in water. Thus, it would be great if author can show or provide more evidences and more details on the cavitation peening in air.

4)      As current single cavitation peening system is only suitable for small surface treatment. Could author provide some application direction to scale the system to apply for large treatment surface or big component?

5)      Author is strongly recommended to provide more technical details as well as methodology as well as give a some recommendations on the development direction and research.   

Reviewer 4 Report

From the formal point of view of the paper I recommend to go through the text and perform language corrections. Here are just some examples of the typing error:

153 –  fur side far side

267 – residuals stress residual stress

352 - too lower downstream pressure  too low downstream pressure

542, 543 - without crack  without crack

and others

I would also recommend to reformulate  

(144 – 145)

expanded region where the pressure increased due to decrease of flow velocity, cavitation collapsed. That means cavitation becomes water.

Especially the term “cavitation becomes water” sounds quite non-standard

Through the text a term “aggressive intensity” is widely cited, however any explanation is not given until page 14. It would be better to include some explanation to the first use of this therm.

Round 2

Reviewer 2 Report

I note improvements in the English.

However, check again the highlighted changes made and ensure the English and spelling are correct.

For example, line 229, p8 onwards, which has both English and spelling errors.

Author Response

Thank you for reviewing my manuscript.

As my manuscript was proofread and corrected by MDPI’s English editing service,

MDPI will check again.

In the case of line 229, p8., I changed as follows.

“In Figure 10, the aspect of cavitation was observed by an instantaneous photography using a flush lamp.”

Reviewer 3 Report

Improve resolution of the figure 1 and 12. Re-plot the figure 15 and 16 for better representation Check grammar and make the correction entire the paper carefully

Author Response

Thank you for reviewing my manuscript.

Comment 1: Improve resolution of the figure 1 and 12.

Answer: I replaced Figs. 1 and 12.

Comment 2: Re-plot the figure 15 and 16 for better representation

Answer: I re-plotted Figs. 15 and 16.

Comment 3: Check grammar and make the correction entire the paper carefully.

Answer: As my manuscript was proofread and corrected by MDPI’s English editing service, MDPI will check again.